# Recent Advances on Targeting Proteases for Antiviral Development

**DOI:** 10.3390/v16030366

**Published:** 2024-02-27

**Authors:** Pedro Henrique Oliveira Borges, Sabrina Baptista Ferreira, Floriano Paes Silva

**Affiliations:** 1Laboratory of Organic Synthesis and Biological Prospecting, Chemistry Institute, Federal University of Rio de Janeiro, Rio de Janeiro 21941-909, Brazil; borges.pho@pos.iq.ufrj.br; 2Laboratory of Experimental and Computational Biochemistry of Drugs, Oswaldo Cruz Institute, Fiocruz, Rio de Janeiro 21040-900, Brazil

**Keywords:** viral proteases, antiviral therapies, peptidomimetics, PROTACs, covalent inhibitors, natural products

## Abstract

Viral proteases are an important target for drug development, since they can modulate vital pathways in viral replication, maturation, assembly and cell entry. With the (re)appearance of several new viruses responsible for causing diseases in humans, like the West Nile virus (WNV) and the recent severe acute respiratory syndrome coronavirus 2 (SARS-CoV-2), understanding the mechanisms behind blocking viral protease’s function is pivotal for the development of new antiviral drugs and therapeutical strategies. Apart from directly inhibiting the target protease, usually by targeting its active site, several new pathways have been explored to impair its activity, such as inducing protein aggregation, targeting allosteric sites or by inducing protein degradation by cellular proteasomes, which can be extremely valuable when considering the emerging drug-resistant strains. In this review, we aim to discuss the recent advances on a broad range of viral proteases inhibitors, therapies and molecular approaches for protein inactivation or degradation, giving an insight on different possible strategies against this important class of antiviral target.

## 1. Introduction

Throughout history, mankind has been afflicted by many infectious diseases, often associated with outbreaks and pandemics with devastating results, leading to economic crisis and disruption of public health systems. Among all of the past pandemics and outbreaks, many have been caused by viral infections, such as the influenza (1918–1919) and swine flu (2009–2010) pandemics, both caused by the H1N1 virus, the 2012 Middle East Respiratory Syndrome (MERS) outbreak, the recent COVID-19 coronavirus pandemic caused by the SARS-CoV-2 and many others [1], which led to millions of deaths worldwide (Figure 1). The spread, and burden, of such diseases have become even greater and more common in the past few decades as we began to live in a highly globalized and interconnected world. Technological advances and economical changes led to increased migration between different countries due to cheaper and faster travels; population growth led to increased contact between people, while climate changes allowed for the existence of pathogens and vectors in places where they were not common before [2,3].

The recent COVID-19 pandemic and the (re)emergence of drug- and vaccine-resistant variants of known diseases has evidenced how quick an infectious disease can spread and the need for different therapeutical approaches to avoid the disastrous consequences of these public health emergences. As most infectious diseases are caused by viruses and given that viruses are prone to mutation for diverse reasons, leading to drug-resistant strains [4,5,6], some effort and attention have turned to them in an attempt to discover different therapeutical approaches.

### 1.1. Virus Replication Cycle

Viruses are microscopic organisms that rely on a host’s cell machinery to reproduce and survive. Their genetic material, which can be composed of either single- or double-stranded [7] DNA or RNA, encodes important proteins necessary for viral replication and maturation. Their genetic material is encapsuled in a protective protein coat, the capsid, with a high structural diversity and can be classified into three general classes, based on their symmetry [8]. The first two classes comprise the helical and icosahedral capsids, while the third includes complex virion structures that do not conform to the aforementioned symmetries, such as seen in poxviruses, geminiviruses and some bacteriophages [9]. In some viruses, the nucleocapsid, i.e., the complex formed of the viral genome and the capsid, can then be surrounded by a lipid membrane and associated proteins which form the viral envelope. This envelope is generally derived from the host cell membrane and contains various proteins that are required for cell recognition and viral entry [10].

Upon infection, the virion attaches to the host cell by interacting with receptor molecules on its surface—proteins, carbohydrates or lipids—in an attempt to cross the plasma membrane to deliver its genetic material into the cytoplasm. In the case of naked viruses, this interaction happens through the viral capsid, while in enveloped viruses it is mediated by entry proteins on the surface of the envelope (e.g., glycoproteins). After binding, a virus uses two main routes to enter the cell: (i) it can gain entry to the cytoplasm by internalization (endocytic route), or (ii) it can directly cross the plasma membrane (non-endocytic route). Some viruses, like the human immunodeficiency virus type 1 (HIV-1) can use both [10,11]. Then, different mechanisms regulate either the fusion of the lipid membrane (in the case of enveloped viruses), or the uncoating of the capsid (in the case of naked viruses), a process that is mediated by many different proteins present in the cytoplasm. Some viruses are also able to directly penetrate the host cell membrane and release its genome [12]. After that, the virus genome and proteins are released into the host’s cytosol, which will then be available for transcription, replication and translation via the host’s cell machinery (not necessarily in that order). The proteins and components produced inside the host cells are then assembled to produce new viruses, which will be ready to infect new cells [7,11] (Figure 2). Several of these steps are facilitated or are highly dependent on a particular class of enzymes, both host’s and viral, called proteases.

### 1.2. Proteases as Targets for Antivirals

Proteases, also known as peptidases or proteinases, are proteolytic enzymes that catalyze the cleavage of proteins or peptide sequences and play a vital role in several biological processes, such as digestion, cell regulation and activity, and diseases and are also used by humans in industrial processes, such as beer and wine clarification and development of drugs [13]. According to the Schechter–Berger nomenclature [14], their active sites are divided into numbered subsites (…S2′, S1′, S1, S2…), where each recognizes specific amino acid residues of the substrate. Then, each substrate’s residue is numbered after the corresponding subsite (P1, P2, P3…), with the P1 position characterizing the position immediately preceding the scissile bond to be cleaved by the protease, as exemplified in Figure 3.

According to the MEROPS Database (https://www.ebi.ac.uk/merops/, accessed on 20 December 2023), proteolytic enzymes can be classified into seven different groups according to the composition of their active site and mechanism of catalysis: aspartic, glutamic, asparagine, threonine, metallo-, cysteine and serine, with these last three being the most abundant in nature [15]. The aspartic, glutamic and metalloproteases rely on activated water molecules as nucleophiles to target and cleave the peptide bond of the substrate, while with cysteine (thiol) and threonine/serine (hydroxyl) proteases the nucleophile is the electronegative heteroatom of the side chain of the residue [16]. Asparagine proteolytic enzymes, however, are named peptide lyases as they do not catalyze any hydrolysis reaction; rather they cleave an intramolecular peptide bond by forming a stable, five-membered succinimide ring through one of its own carbonyl groups from the backbone and an amide group of the side chain [17] (Figure 4).

One important aspect of the replication strategy adopted by many viruses is the encoding of a large precursor polyprotein into the genome, which confers many benefits such as condensation of genetic material and regulation of protein activity [18]. These polyproteins can be cleaved either by the host’s cellular proteases or through an embedded protease, which is responsible for the self-cleavage in specific sites leading to the generation of non-structural proteins (nsp) and mature viral proteins that are necessary for viral maturation and assembly. This strategy allows for a more compact genome and eliminates extra cellular machinery to process and express each required protein [19]. Thus, it becomes clear that viral proteases play an important role in replication and maturation, making them an attractive molecular target for the development of new drugs.

So far, many approaches have been developed to target viral proteases, some with more success than others. As we shall discuss throughout this review, they can include direct inhibition of the active site of the target protease, by exploring the chemistry behind the cleaving of the peptide bonds, or through other regions of the protein, the allosteric sites, taking advantage of the fine-tuned control that such sites have on protein regulation. Others can even rely on the inhibition of the host’s proteases, which plays a key role in viral entry and internalization. With the emergence of drug-resistant viral strains and the ever-growing risk of new viral outbreaks, developing new strategies to target this important enzyme class has become even more important. Thus, in this review, our aim is to discuss the mechanisms behind the roles of proteases on the viral life cycle and we shall cover the different approaches that have been recently considered to develop new and innovative antiviral therapies. We hope that our discussion can bring light to this important matter and aid in the development of new and innovative ideas for drug discovery.

## 2. Inhibition of Viral Proteases by Targeting the Active Site

### 2.1. Peptide-Based Molecules

Perhaps the most common strategy behind protease inhibition relies on the use of low molecular weight molecules that target the orthosteric site of the enzyme in a competitive manner, usually achieved through the design of molecules that are similar to the substrate of the target protease [20]. These inhibitors are designed to have higher affinity towards the protease’s subsites than its natural substrate while mimicking the transition state of the enzyme catalyzed reaction, effectively blocking the substrate’s access to the active site and impairing its activity [21].

Since proteases cleave peptide bonds, most researched protease inhibitors are peptide-based molecules, which include the use of naturally occurring amino acids, or peptidomimetics, which contain modified amino acids with enhanced molecular properties that increase their affinity towards the active site, while also improving their pharmacological properties by enhancing stability [22]. According to El-Faham et al. [23], peptide-based molecules account for 5% of the total number of drugs approved by the FDA between January 2015 and May 2021, with 21 drugs being launched during the same period for many targets. More recently, in 2022, of the 37 FDA-approved drugs, five belong to the TIDES (oligonucleotides and peptides) category, accounting for almost 15% of the total approved drugs [24,25].

The first protease inhibitor, saquinavir (saquinavir mesylate, **1**), was developed in 1990 [26] and approved by the FDA for clinical use in 1995 as an anti-HIV drug, setting the ground for subsequent analogues [27] and initiating an era of protease inhibitors [28,29]. Today, against HIV alone there are ten FDA-approved protease inhibitors, while other diseases, such as hypertension, AIDS and cancer, have also benefited by the discovery of inhibitors [20]. The hepatitis C virus (HCV) is another major infection with an enormous global health burden with no available vaccine, affecting 58 million people worldwide and causing 290,000 deaths annually [30]. The most notorious HCV drugs, boceprevir **9** and telaprevir **8**, are also peptidomimetic molecules, with the most recent drugs following the same trend [31] (Table 1).

Today, peptide-based molecules, including peptidomimetics, are expected to see an increase in market expansion, reaching a value of USD 46.6 million by the end of 2024 [33]. Together with the emergence of several viral infections in the last years and the elucidation of their protease structures, as well as with advances in artificial intelligence and tools like Alphafold [34,35], development of such inhibitors has gained some traction. A successful example is the recent approved drug Paxlovid^®^ (nirmatrelvir/ritonavir tablets, co-packaged, **12**) to treat the COVID-19 disease. Nirmatrelvir is a peptidomimetic molecule containing a nitrile group as its warhead (see Section 2.2 below), an electrophilic moiety designed to promote the formation of a covalent bond with the catalytic residue of the active site (Figure 5). This molecule acts by inhibiting the SARS-CoV-2 main protease (M^pro^), an important protease responsible for the viral replication inside cells, through the formation of a covalent bond with the catalytic cysteine [36], rendering the protease incapable of processing the precursor polyproteins encoded into the viral genome [37]. Paxlovid^®^ was approved by the FDA in May 2023 for the treatment of mild-to-moderate COVID-19 in adults who might be at risk for progression to severe COVID-19 [38].

Peptide-based and peptidomimetic molecules have been considered for many different viral infections as potential protease inhibitors. Some typical and often researched infections are the Zika (ZIKV) and Dengue (DENV) virus. Zika and dengue are both mosquito-borne pathogens and belong to the same *Flaviridae* family, together with West Nile virus, yellow fever and Japanese encephalitis virus [39,40]. Both ZIKV and DENV are common in tropical countries and are still a threat in low and middle-income countries, such as Brazil and India, and in some regions of Africa and Asia [41,42]. Their genomes are composed of a single-stranded RNA, encoding a single polyprotein that acts as a precursor to other viral proteins. A two-component viral protease, NS2B-NS3, is a serine protease formed from the non-structural protein NS3 and the NS2B co-factor, that cleaves the precursor polyprotein into individual proteins required for viral replication [43,44]. This protease belongs to the trypsin superfamily with a preference for substrates with dibasic residues, such as arginine and lysine, and its active site contains a highly conserved catalytic triad composed of His51, Asp75 and Ser135 residues [45,46,47].

Although there is a vaccine for DENV, there is no specific treatment for ZIKV so far, and drug-resistant strains are always a concern regarding viral infections. So, ongoing research has focused on developing inhibitors of NS2B-NS3, with several being peptidomimetic. In 2018, Phoo et al. obtained high-resolution crystal structures (PDB ID: 5ZMQ, 5ZMS and 5ZOB) of the unlinked NS2B-NS3 protease (bZiPro) from ZIKV with three peptidomimetic ligands originally designed as WNV protease inhibitors [48]. Compounds **13**–**15** (Figure 6) contain a specific P1–P4 segment with different P1′ residues that can be cleaved by bZiPro, which can be detected via mass spectrometry. Their findings have shown that the bulkier P1′ amidinobenzylamide moiety present in compound **15** turns the P1–P1′ amide bond more resistant to cleavage, as shown via mass spectrometry and crystal data, which led to a more potent inhibitor.

The stability of the amide bond surrogates is an important factor when developing peptidomimetic inhibitors, since it can have a direct impact not only on inhibitory activity, for example as seen for **15**, but more importantly on the bioavailability of such compounds, by preventing degradation by host peptidases [49]. In a more recent study, Behrouz et al. designed potent peptidomimetic inhibitors containing a sulfonyl moiety as N-terminal cap [50]. Sulfonamides are important structural motifs that can be regarded as surrogates that mimic peptide bonds while maintaining an increased bioavailability and stability towards proteolytic degradation [51]. Indeed, the authors have observed that replacement of the amide group with a sulfonamide at the N-terminal of DENV protease inhibitors has shown an improved metabolic stability against rat liver microsomes, although some loss in activity was observed. Nevertheless, it was observed that the presence of the sulfonamide groups has also improved stability and off-target selectivity against pancreatic serine proteases, trypsin and α-chymotrypsin, and thrombin [50].

Such chemical modifications on a lead structure are a widely known strategy to enhance its pharmacokinetic properties, leading to higher bioavailability or stronger interactions with the target protease. An interesting approach has been the development of macrocyclic surrogates to increase the compounds’ affinity towards featureless or shallow active sites. This is better exemplified by the development over the years of macrocyclic peptidomimetic or peptide-based inhibitors of the HCV NS3/4A serine protease. Its active site is located on the surface of the protein and comprises a catalytic triad of His57-Asp81-Ser139 and contains a shallow, nonpolar and highly solvent-exposed specificity pocket, which makes it difficult to develop potent inhibitors [52,53]. The first macrocyclic HCV NS3 inhibitor to be approved by the FDA was simeprevir (Table 1, **11**), followed by the approval of grazoprevir **16**, glecaprevir **17** and voxilaprevir **18** [54], and since then the macrocyclization strategy has been explored for other proteases as well. It has been shown that macrocyclization allows the selective binding of compounds on relatively shallow protein surfaces involved in clinically important protein–protein interactions (PPIs), while also providing conformational restraints that result in minimal entropic loss due to binding, strongly enhancing the inhibitors’ potency [55,56]. Following this trend, Nageswara Rao et al. have developed a structure-guided strategy to design two series of compounds with different quinoxaline moieties at position P2 and diverse P4 groups to avoid drug resistances that could emerge from the structurally similar quinoxaline-based compounds **16**–**18** [57]. Their findings have shown that 3-methylquinoxaline derivatives on P2 and fluorinated groups at the P4 cap are able to improve the observed potency profiles on resistant strains of HCV, ultimately leading to the development of compounds **19**–**22** (Figure 7).

Despite the success of such compounds and several advances in the field, peptide-based and peptidomimetic compounds still have several drawbacks, specifically regarding their pharmacokinetic profile. Their relatively high molecular weight limits their absorption, specially by the blood–brain barrier (BBB), which limits their use on central nervous system (CNS) targets. Being structurally based on peptides, it also means they can be degraded by other peptidases, lowering their overall bioavailability, or interact with off-targets, resulting in adverse effects [58]. On the other hand, targeting the catalytic residues of proteases has its own challenges, since viral mutation can induce structural changes in the active site, rendering electrophilic warheads useless over time and resulting in loss of selectivity towards the overall structure of the inhibitor. Thus, other strategies have been sought to overcome such challenges and to explore new and innovative ways of targeting viral proteases.

### 2.2. Covalent Inhibitors

Besides stability, the selectivity and potency of peptide-based and peptidomimetic compounds can be improved by exploring the cleavage mechanisms of proteases, specifically by targeting the nucleophilic residues in the active site. Normally, small molecules interact in a fast and reversible manner with the target protein, maintaining an equilibrium between the unbound protein and the protein–ligand complex, which ultimately can result in lower response. However, in some cases, prolongation of the therapeutic response is desired, which can be achieved by increasing the time available for this small molecule to interact with the protein’s active site [59]. This can be done by promoting the formation of a reversible or irreversible covalent bond between the target protein and the ligand, which can be accomplished using electrophilic groups called warheads [59,60]. Today, many warhead groups have been studied for different targets, such as nitriles [61,62], alkynes [63], aldehydes [64,65], unsaturated α,β-carbonyl [66,67,68], alkyl halides [69] and many others [70,71,72] (Figure 8).

The choice of warhead groups is crucial in the development of viral protease inhibitors and, more broadly, in the design of any protease inhibitor. Understanding their reactivities towards specific residues is key to promoting either reversible or irreversible bindings, which can result either in a potent or a non-selective inhibitor. This is because the irreversible covalent bond can result in longer residence times, which can happen either on the target viral protease, with an observed therapeutical effect, or on a host’s proteases, resulting in several adverse reactions and toxicity [73].

To better understand this, Müller et al. led a study that aimed to investigate the influences of different known warhead groups and different peptide sequences on selectivity and reactivity on different proteases [74]. They synthesized different compounds containing specific peptide sequences coupled with different warhead groups, which were tested in vitro against five different proteins (uPA, CatS, β5-subunit of the proteasome, SARS-CoV-2 M^pro^ and rhodesain), representing three groups of proteases: serine, cysteine and threonine. These compounds were also tested against model nucleophiles to probe their reactivity in a system prepared to mimic the catalytic residues. Their findings have shown that, indeed, specific peptide sequences are key to producing more selective inhibitors, with some compounds only showing inhibitory activity when built with the suited sequence, while as for the warhead groups, the difference in its electrophilic profile plays an important role in defining the reactivity, and in some cases the affinity, of the target compound towards the protease. Investigational studies like these [74,75] show the importance of the rational design of peptide-based and peptidomimetic protease inhibitors, and highlight that it needs to happen in two distinct stages: (i) interaction of the inhibitor with the target protease via non-covalent bonds, which depends on the overall structure of the ligand and the active site, and (ii) reaction of the ligand with the catalytic residue via covalent bond formation [76], which will depend on the warhead group. Increasing the reactivity of the warhead group can bring some advantages like lowering the dose needed to produce a therapeutic effect and being able to target shallow binding sites, but it does not translate to a failproof approach, as it can affect its selectivity which leads to toxicity [59]. Thus, some approaches have been developed to attenuate such effects.

For example, the inherent electrophilicity of certain warhead groups, such as aldehydes, often translates into high inhibitory potency, but it also contributes to poor pharmacokinetic profiles and high toxicity [77]. Thus, one recent strategy is to “mask” such groups, effectively turning them into prodrugs, so they can be converted into their active forms via metabolic reactions [78,79]. For example, Li et al. have developed a novel class of self-masked aldehyde inhibitors (SMAIs) to target cysteine proteases, such as cruzain, from the parasitic protozoa *Trypanosoma cruzi* [80]. They modified the P1 group of a peptidomimetic inhibitor into an ortho-tyrosine, allowing for a close proximity of the hydroxyl group to the carbonyl, resulting in the formation of a *δ*-lactol group, which remains stable in aqueous solution, while the open form is only achieved when bound to the protease. They also suggest that the same strategy could be applied to other cysteine proteases, like SARS-CoV-2 M^pro^, which led to the bioisosterical replacement of the P1 γ-lactam ring to the 2-pyridone moiety (Figure 9). Other novel warhead groups and strategies have also been proposed, such as switchable electrophiles, strain-release electrophiles, cross-linking warheads, and many others [81]. Such strategies might be an interesting approach when designing specific inhibitors to a target protease, while also avoiding toxicity profiles.

### 2.3. In Silico-Designed Inhibitors

Another interesting strategy in the development of new protease inhibitors, and drugs in general, is the use of Computer-Aided Drug Design (CADD) or in silico tools that aim to speed up and lessen the cost of the dispendious drug-discovery process. These tools can range from the prediction of the binding mode of a particular set of molecules in a structure-based drug design (SBDD), using molecular docking and molecular dynamics simulation, to the virtual screening of millions of molecules in a ligand-based drug design (LBDD) fashion, or through the use of artificial intelligence (AI) and machine learning (ML) models [82].

In SBDD approaches, previous knowledge of the molecular target is needed, where previous reported structural data, usually via x-ray crystallography [48,83,84,85], allows for the characterization of the active site of the protease with different co-crystallized ligands. This information is vital in the development of new drugs since it can shed some light behind the important interactions between inhibitors and target protease. These results can then be used in computational efforts to evaluate hundreds or thousands of compounds in a short time and aid in the development of more potent inhibitors. For example, Shin et al. performed a virtual screening using 467,000 structurally diverse chemical compounds against ZIKV NS2B-NS3 protease (PDB ID 5H4I). From 123 candidates, they found that **22** had the highest potency, reaching an IC_50_ value in the sub-micromolar range [86] (Figure 10). A similar work published by Hossain et al. depicts a virtual screening of 429 antiviral peptides against the same target. Molecular docking and molecular dynamics were performed to evaluate the interactions of known antiviral peptide-based compounds against a four-amino acid peptide as control. Their analysis suggests that four peptide inhibitors are promising as NS2B-NS3 inhibitors [87]. Similar computational works recently published are available for both ZIKV and DENV [88,89].

The field of AI has seen an increase in drug research over the past few years due to the availability of an immense amount of data and increased computational power that is readily available. In short, AI involves several advanced tools, such as artificial neural networks (ANNs) and other deep learning (DL) models that aim to mimic human intelligence, by interpreting and learning from any given data how to make decisions for any specific objective [90]. One of the advantages of such tools in drug R&D is their ability to handle large volumes of data that can be used to predict and suggest new and potentially active molecules [91].

For example, Saramago et al. have identified three potent molecules with anti-SARS-CoV-2 activity through an AI-driven approach followed by in vitro validation [92]. This was achieved by training a recurrent neural network (RNN) to design textual representations of molecules through the simplified molecular-input line-entry system (SMILES) and then fine-tuning the model towards the generation of fragment-like molecules with potential activity against SARS-CoV-2 M^pro^. Out of the 37 obtained fragments, five (compounds **23**–**27**) have shown activity against M^pro^, with one of the compounds, a chloromethyl amide derivative of 8-methyl-gamma-carboline **23**, reaching an IC_50_ of 1.51 μM (Figure 10). Despite the chloromethyl amide being a known electrophilic warhead, reversibility assays have shown that this compound acts as a reversible inhibitor of M^pro^.

In another paper, Arrigoni et al. used two neural networks that worked in tandem to search for HIV-1 protease inhibitors in a database containing 250,000 molecules that were further assessed through molecular docking and molecular dynamics simulation [93]. Their model consisted of a variational autoencoder (VAE), which was responsible for generating a numerical representation of the input molecules to be used in the next neural network, a deep feed-forward network (DNN), which associates the numerical representation with the efficacy of known ligands of the target protease. Their findings suggest that one molecule, ZINC991374169, could be potentially active against HIV-1 protease. Despite not being experimentally validated, the computational protocol used in this work is simple and does not require large computational power, demonstrating the potential of AI techniques to greatly accelerate virtual screening processes. Both findings highlight the potential of AI in accelerating the discovery protease inhibitors, which is crucial in addressing the challenge of drug resistance and the dispendious drug-discovery process.

## 3. Inhibition of Viral Proteases through Allosteric Site Modulation

Allosterism, or allostery, is a well-known concept in molecular biology and an important phenomenon in protein regulation. It is defined by the binding of an effector, i.e., a small molecule or another protein, at a region that is topologically distinct from its functional site. This leads to a conformational change due to a series of physical strains in the protein, ultimately altering the properties of its functional site, which can modulate either negatively or positively its biological activity. Thermodynamically, this means the disturbance of the equilibria between populations in the “active” and “inactive” conformation by stabilizing one of these forms. Thus, allostery produces a fine-tuned control that plays a key role in signal transduction, metabolism regulation, enzyme activation, motor work and transcription control [94], which have been explored in drug discovery [95].

Differently from their orthosteric counterparts, allosteric inhibitors act in a non-competitive manner by not targeting the protease’s active site. Rather, their main goal is to disturb the active/inactive conformational equilibria by targeting another region of the protein, effectively acting as “molecular switches”, forcing the protease into an inactive conformation (Figure 11). It has been found that dihydroxy-substituted benzothiazole moieties are an important unit that are associated with non-competitive inhibition attributed to the binding at an allosteric site of the DENV NS2B-NS3 serine protease [96]. Due to the poor water solubility profile of such molecules, Millies et al. further developed these compounds by exchanging the previous ortho-substituted aromatic ring system via a proline, increasing its solubility as allosteric inhibitors within the sub-micromolar range [97]. They suggested, through molecular docking and site-directed mutagenesis, that the compounds inhibit the protease through a non-competitive mechanism by binding at an allosteric site and interacting with Asn152, a key residue that seems to be responsible for shifting protein conformations, and effectively stabilizing the protein in the open and inactive conformation. Since the NS2B-NS3 allosteric sites are highly conserved among ZIKV and DENV, such compounds could be considered for both diseases. A computational study carried out by Gangopadhyay et al. has shown that polyhydroxy compounds, like flavonoids and polyphenols, are also potential allosteric inhibitors of NS2B-NS3 [98], which relates to the previous results seen for dihydroxy-substituted benzothiazole structures.

Some viral proteases are only active when in the form of homodimers, which promotes the ordering of the active site for substrate recognition [99,100,101,102]. It is important to point out that the dimerization process can occur in various regions, including allosteric sites. In the case of proteases, allosteric regulation may influence dimerization, substrate binding or other aspects of enzyme function. In fact, dimerization disruption has been shown to be an interesting approach with the approval of the HIV-1 protease inhibitor, darunavir (DRV), by the FDA in 2006 for treatment of patients infected with drug-resistant variants. Due to its unique structure design, DRV impairs the protease activity in a unique dual mode of action; (i) it inhibits the catalytic dimer and (ii) it inhibits monomer dimerization and formation of the active enzyme by promoting extensive interactions throughout the protease active site. Moreover, DRV has also displayed broad-spectrum antiviral activity against multidrug-resistant variants of HIV-1 [103]. Thus, inhibition of the dimerization process is an interesting approach for the design of protease inhibitors.

For example, it has been shown that the SARS-CoV-2 M^pro^ is only active when in the form of a homodimer, with mutations at the dimer interface heavily impacting on overall protease activity [104,105]. With that in mind, Silvestrini et al. decided to analyze SARS-CoV-2 M^pro^ dimerization dynamics through Small Angle X-ray Scattering (SAXS) for more accurate insights, with and without inhibitors, while also assessing their dimerization inhibition activity [106]. Their study provides a detailed thermodynamic picture of the monomer–dimer equilibrium and investigates how the M^pro^ dissociation process is affected by small inhibitors selected by virtual screening. The inhibitors were found to affect dimerization and enzymatic activity to a different extent and sometimes in an opposite way, likely due to the different molecular mechanisms underlying the two processes. For example, since the N-finger Ser1 of one monomer and Glu166 of the other participates in the shaping of the active site [107], interactions with these residues could disrupt the dimerization equilibria. In another paper, Tao et al. found that metallodrugs, such as colloidal bismuth citrate, could inhibit M^pro^ by binding to Cys300, an important residue that stabilizes at the dimerization domain and disrupts the monomer–dimer equilibria [108]. It has also been suggested that niclosamide derivatives can inhibit Mpro via allosteric inhibition, although the mechanism has not been elucidated yet [109].

Hulce et al. synthesized a series of human herpesvirus (HHV) protease inhibitors that act by covalently binding to a non-catalytic residue [110]. The HHV protease is a family of structurally conserved serine proteases where the active site is comprised of a non-canonical His-His-Ser catalytic triad. The catalytic serine residue, however, is not a strong nucleophile, which creates a challenge when developing electrophilic warheads [111]. Thus, an interesting approach is to target a more nucleophilic residue that might produce a therapeutical effect. One of the inhibitors, compound **28**, promotes protein dimerization by covalently binding to a non-catalytic cysteine (Cys161), however in a non-productive state that resulted in loss of activity (Figure 12). This is explained by the disruption of the active site promoted by the binding, as seen via molecular dynamics and X-ray crystallography results. Since dimerization is a common phenomenon within viral proteases [99,100,101], this same strategy can be expanded to other viruses, such as SARS-CoV-2, West Nile Virus (WNV), Zika and HIV, especially if protonation states are known [112].

Inhibiting proteases via allosteric sites offers some quite interesting advantages over orthosteric inhibitors; however, there are some caveats that must be addressed before considering this approach. While it can offer some selectivity since they do not bind to the highly conserved active sites, developing such inhibitors is challenging due to a lack of knowledge about the allosteric site location [113]. Although it is possible to imply its location through computational approaches, the lack of experimental data makes the development of such inhibitors quite demanding. Moreover, while allosteric inhibition may allow for modulation of the protease’s activity, rather than impairing it, it can also produce detrimental effects by activating the protease and developing the disease [114]. Such an approach might be beneficial when considering drug-resistant strains or mutants, in which the active site can no longer be targeted by known inhibitors, or when a fine control of the protease activity is needed. Nonetheless, allostery is a quite complex mechanism [115], but it can also be a promising alternative approach in drug discovery.

## 4. Host Proteases as Potential Drug Targets for Antiviral Therapies

Many viruses exploit host proteases during their replication cycle. For instance, some viruses use host proteases to facilitate their entry into host cells. For example, the influenza virus has a surface glycoprotein called hemagglutinin that requires cleavage by host proteases in a process known as “priming”, an essential step for the fusion of the viral and host cell membranes to become active [116,117,118,119] (Figure 13). Despite not being of viral nature, host proteases can also be an interesting target for the development of antiviral therapies. For example, human cathepsin L (CatL) is a lysosomal cysteine protease and has been identified as a target that some viruses, such as coronaviruses, may exploit for entry into host cells [120,121]. In the context of viral infections, cathepsin L can activate viral fusion proteins, facilitating the entry of the virus into the host cell.

In the case of the SARS-CoV-2 coronavirus, after internalization through endocytosis the CatL catalyzes the cleavage of the spike protein (S) on its surface, allowing for the release of the viral RNA [122]. Similarly, it has been established that SARS-CoV-2, especially the Beta and the globally spread Omicron variant, also uses the transmembrane serine protease 2 (TMPRSS2) for viral entry into lung cells, proving to be an essential target for therapeutical intervention [123,124]. By exploring host proteases, Mellot et al. have shown that the previous antiparasitic agent, K777 **29**, could block SARS-CoV-2 entry in host cells [125] (Figure 13). **29** is a peptidomimetic containing a vinyl sulfone warhead and a known CatL and cysteine protease inhibitor. The authors have found that **29** is a selective inhibitor of CatL, with higher inactivation rates of the latter than cathepsins B and K, and it inhibits viral entry in four host cells models (Vero E6, HeLa/ACE2, A549/ACE2 and Calu-3/2B4) with nanomolar potency. Following these results, Zhu et al. aimed to develop a series of SMAIs that could inhibit both CatL and SARS-CoV-2 M^pro^ [126] by finding a unique peptide scaffold that would be recognized by both proteases. Although none of the compounds were able to inhibit M^pro^ directly, they found that some of the SMAIs were able to produce an inhibition of the cytopathic effect on Vero E6 and A549/ACE2 cells infected with SARS-CoV-2, indicating that a bi-functional strategy might be feasible with proper optimization, while also indicating that targeting the host’s proteases is also a possible strategy to prevent viral infection.

The complex mechanism and the role of the host’s cell proteases behind viral entry can provide many different approaches. Several other host cell proteases that mediate viral entry have been identified as potential targets for drug intervention, such as furin [127], cathepsin B, trypsin and Factor Xa [128]. A thorough review has also been recently published [129] depicting many different proteases that regulate viral entry for several viral families, which can provide several targets for drug development. Some innovative approaches can also be suggested. It has been observed that viral entry inhibition is only possible in the initial stages of infection [130]; thus, one can envision the application of such a strategy in combination therapies that target different stages of infection, possibly potentializing the therapeutical effect. Of course, the host’s cell membrane protein might participate in many important biological processes and modulating their activity could also result in detrimental effects.

## 5. Targeted Protein Degradation Strategies

In the dynamic landscape of cellular processes, the regulation of protein levels is a fundamental aspect of maintaining cellular homeostasis and orchestrating various physiological functions. One of the key mechanisms governing protein turnover is ubiquitination, a tightly controlled post-translational modification. Ubiquitination involves the covalent attachment of ubiquitin (Ub) molecules to target proteins, marking them for degradation by the proteasome or modulating their activity [131,132,133]. For example, in K63 ubiquitination, Ub molecules are attached to the Lys63 (K63) residue of the target protein, marking it for different signaling pathways, with important roles in DNA damage repair, histone regulation, signal transduction and autophagy, while in K48 ubiquitination the polyubiquitin chain marks the protein for proteasomal degradation [134]. The proteasome itself is a large protein complex that consists of two major portions: a catalytic 20S core particle of nearly 700 kDa, and one or two associated 19S regulatory particles of nearly 900 kDa. The 20S proteasome is a cylindrical particle formed by four heteroheptameric rings (α and β), each consisting of similar α- and β-subunits, with some of the β-subunits having proteolytic activity (Figure 14) [135]. While polyubiquitination can result in protein degradation by the proteasome, effectively leading to the inactivation of the protein, proteasome inhibition can also lead to accumulation of polyubiquitinated proteins, resulting in cell stress response and death [15]. The exploration of such an intricate mechanism has made possible the development of molecules that can interfere with this pathway and modulate key targets to produce a therapeutical effect [136].

For example, it has been shown that ZIKV infection is highly dependent on polyubiquitination of the envelope protein (E) by the E3 ubiquitin ligase TRIM7 through K63-linked polyubiquitination, leading to an increase in the viral attachment to host cell receptors and viral entry [137]. More recently, Wang et al. discovered a new potential target against ZIKV infection, ubiquitin-specific peptidase 38 (USP38) [138]. USP38 is a member of the ubiquitin-specific processing enzyme family, and its functions have been associated with several diseases, such as asthma and cancer [139,140], and with drug resistance [139]. The authors have shown that USP38 acts by decreasing the polyubiquitination on K63, effectively impairing viral entry on host cells. So far, to the best of our knowledge, there is no inhibitor that makes use of this pathway, suggesting that USP38 is a potential novel target against ZIKV infection.

In another example, by screening NS2B3 protease inhibitors for both ZIKV and DENV, Ci et al. found that bortezomib **30**, a proteasome inhibitor used to treat multiple myeloma and mantle cell lymphoma [141], could induce NS3 ubiquitination and aggregation, lowering the activity of the protease inside cells [142]. NS2AB3 is a fusion protein that is self-cleaved at specific sites to produce nonstructural proteins (nsp) which reflects the protease activity of NS2B3. When assessing the ability of five proteasome inhibitors (bortezomib **30**, carfilzomib **31**, ixazomib **32**, epoxomicin **33** and oprozomib **34**) (Figure 15) to impair NS2AB3 self-cleavage, the authors have observed that all five drugs were able to inhibit viral replication in BHK-21-infected cells, but surprisingly none of them were able to inhibit the NS2B3 protease, meaning that another pathway was involved. Ultimately, they found that **30** impaired NS2AB3 cleavage by inducing NS3 ubiquitination and aggregation through the modulation of two key E3 ligases from the endoplasmic reticulum-associated degradation (ERAD) pathway. These data suggest that the ERAD pathway not only plays a significant role in ZIKV/DENV replication, but also indicates a novel approach for antiviral therapy.

While ubiquitination is a well-established regulatory pathway, recent innovations in drug development have given rise to a revolutionary class of compounds known as PROTACs, or proteolysis-targeting chimeras, as a new targeted protein degradation (TPD) strategy [143]. These molecules are heterobifunctional small molecules consisting of two ligands joined together through a linker. One ligand is responsible for binding to a protein of interest (POI), while the other is responsible for recruiting and binding to an E3 ubiquitin ligase. Because of the linker, the two proteins are positioned next to each other and promote polyubiquitination of the POI, ultimately leading to the proteasomal degradation of the POI [143,144] (Figure 16). Thus, PROTACs exploit the ubiquitin-proteasome system to selectively eliminate specific proteins, opening new avenues for therapeutic interventions. A rise in PROTAC research has been seen mostly on cancer research [143,145], but it has also been explored for antiviral therapy.

Based on this premise, it is possible to envision a strategy correlating PROTACs and known inhibitors. As such, indomethacin (INM) **35** is a non-steroidal anti-inflammatory drug (NSAID) that has shown to have activity against SARS-CoV-2, although its mechanism is yet unknown [146,147]. Accordingly, Desantis et al. have explored this interaction to develop INM-based PROTACs linked to the Von Hippel Lindau (VHL) **36** E3 ligase ligand and evaluate their antiviral activities against several coronaviruses [148]. Their results showed that the developed PROTACs demonstrated up to fivefold improved ability in inhibiting viral replication compared to indomethacin alone. Furthermore, two INM-based PROTACs, **37** and **38**, exhibited specific antiviral activity against pandemic and epidemic coronaviruses belonging to different genera of the Coronaviridae family, suggesting a broad-spectrum property as pan-coronavirus agents (Figure 17). In a similar study, Alugubelli et al. proposed other PROTACs based on known peptidomimetc M^pro^ inhibitors, MPI8 **40** and MPI29 **41**, linked to the ubiquitin E3 ligase cereblon (CRBN) ligand, thalidomide **39**, as the E3 ligase binders [149]. All the developed compounds, **43** and **44**, exhibited sub-micromolar IC_50_ values, underscoring the potential of these PROTAC degraders as a novel approach for targeting SARS-CoV-2 and inhibiting viral replication (Figure 18). It is interesting to note that both strategies have explored different approaches on the same target. While the first have explored a known activity of a compound without its mechanism, the latter had the knowledge of the target’s structure, allowing for ligand optimization. Yet, both strategies proved successful. This phenomenon is well correlated with the many advantages seen for PROTACs, including the ability to target proteins and induce degradation even with modest affinity ligands, allowing for the targeting of proteins that were previously considered difficult to drug or with no known mechanism [150].

Hahn et al. analyzed the efficacy of the commercially available cyclin-dependant kinase 9 (CDK9)-directed PROTAC, THAL-SNS032 **45**, as a potential human cytomegalovirus (HCMV) antiviral [151]. CDK is a type of protein kinase involved in the regulation of the cell cycle, transcription, metabolism and apoptosis. It plays a crucial role in controlling the progression of the cell cycle by phosphorylating specific target proteins and has been found to be involved in the progression of many diseases [152]. HCMV has been shown to interact with the host cell cycle machinery, including CDKs, to create a favorable environment for viral replication [153]. Since human CDK inhibitors have previously been shown to have anti-cytomegaloviral properties [154,155], the authors have decided to assess the antiviral properties of **45**, which is comprised of a link between **39** and the protein kinase inhibitor SNS032 **42**, towards HCMV (Figure 18). The authors have demonstrated that **45** induces the degradation of CDKs (1, 2, 7 and 9) and has shown a 3.7-fold stronger inhibition effect when compared to **42** alone.

All the above examples show the current landscape of PROTAC-based therapy and how this new strategy can be an improvement over current drugs. In the last few years, we have seen some promising results with ongoing clinical trials of two PROTACs, ARV-110 and ARV-471, with others expecting to enter phase I in the near future [144]. However, it is important to note that PROTACs are a relatively new class of drugs, and their safety and efficacy in humans are still being evaluated. Additionally, the development of PROTACs requires a deep understanding of the target protein and its interactions, which can be challenging for some viral proteins. Furthermore, the use of PROTACs may require the identification of specific E3 ligases that can recruit the target protein for degradation, which may not always be feasible. Despite these challenges, some alternatives have been proposed, such as bioPROTACs and hybrid PROTACs, that are yet to be fully explored [144]. Overall, while PROTACs show great promise as a potential treatment for several diseases, further research is needed to fully understand their limitations and potential side effects.

## 6. Natural Compounds as a Source of Novel Antiviral Drugs

Natural products are an interesting and often overlooked source of bioactive compounds due to their intrinsic chemical and structural diversity, which has yielded many potential drugs today, such as morphine, quinin and penicillin, with several others being studied for their antiviral properties [156]. Due to their privileged scaffolds and promiscuous binding nature [157], natural compounds do not usually have a “preferred binding mode” or a specific mechanism of action, which made us consider it a separate section on its own.

Green tea (*Camellia sinensis*) is regarded as the second most popular drink in the world and is extensively researched, and used, for its pharmacologically active components, including anti-inflammatory properties, and remedial effects for diabetes, Alzheimer’s disease, oral cancer and dermatitis [158]. Recently, Coronado et al. identified that epigallocatechin **46** and epigallocatechin gallate **47**, natural occurring polyphenols present in green tea, and their derivative EGCG octaacetate **48** (Figure 19) are able to inhibit flaviviruse proteases within the sub-micromolar range [159]. Surprisingly, although all compounds share a common scaffold, they have exhibited different binding modes with **46** exhibiting a competitive inhibition behavior, while **47** and **48** are non-competitive inhibitors, possibly acting on an allosteric site.

The SARS-CoV-2 papain-like protease (PL^pro^) is another cysteine protease that processes the released polyproteins, pp1a and pp1ab, into mature nsps and thus has a pivotal role in viral replication and controlling host cell response [160]. Besides cleaving the polyprotein, PLpro also removes ubiquitin and Interferon-stimulated gene 15 (ISG15) from host-cell proteins in an attempt to evade the host’s innate immune responses [161]. Srinivasan et al. identified three natural phenolic compounds, 4-(2-hydroxyethyl)phenol **49**, 4-hydroxybenzaldehyde **50** and methyl 3, 4-dihydroxybenzoate **51**, as potential inhibitors of PL^pro^ (Figure 19) [162]. These compounds were found to bind at an allosteric ISG15/Ub-S2 binding site in the thumb region of PL^pro^, leading to the inhibition of PL^pro^ in deISGylation activity assays. Importantly, the binding of these compounds disrupted essential molecular interactions with ISG15, which is crucial for the virus to evade the host’s innate immune responses. Furthermore, the study suggests that these compounds could serve as promising lead compounds for the development of specific coronaviral PL^pro^ inhibitors by exploring two different mechanisms: allosteric targeting and ubiquitination modulation.

It is worth noting that natural products are not a proper “mechanism” to be explored per se, but rather a valid strategy for drug discovery. While the mechanisms behind their biological activity may not be evident at first, we have discussed examples in which these molecules can lead to surprisingly effective applications and the discovery of potential therapeutical targets.

## 7. Combination Therapies and Synergistic Approaches

Though we have focused on individual molecular targets or approaches so far, it is pertinent to highlight that another possibility arises when these are used together in combination therapies by exploring possible synergistic effects. In fact, it has been shown that targeting different stages of the viral replication cycle often leads to an improvement in treatment of the viral infection [163] and such a strategy has been explored for different diseases, such as cancer and hypertension [164]. For example, it is possible to combine viral entry inhibitors and direct acting antivirals to achieve a strong synergistic effect that is on par with current antiviral therapies [165]. Another possibility is to target similar proteins that are present both in humans and viruses [166]. This strategy can also come with additional positive effects, like preventing the development of drug-resistant strains, since mutations on a host’s cell proteins usually does not occur, and reduced doses for viral treatment due to higher therapeutical response [164]. Though this approach is outside the scope of this review, we would like to shed some light on this matter as another possibility for the development of potential antiviral treatments.

## 8. Conclusions

The mechanism of viral infection, from its entry to replication, is an intricate dance that involves multiple pathways, both viral and host’s. Though it might seem challenging, this myriad of pathways also offers multiple potential targets that can be explored in drug development. So far, the attention has been mostly directed to viral targets that could directly inhibit replication, such as viral proteases, a trend that can be seen in most recent developments for different diseases [167,168]. However, viruses are highly prone to mutation, quickly rendering most of the researched targets ineffective within a short period of time and requiring new approaches on a regular basis. This is especially true considering the quick emergence of viral outbreaks and pandemics. Also, drug-resistant strains pose other challenges when considering chronic and acute infections, as they dictate not only the success of such antiviral treatments, but also the treatment protocol itself [169].

As we have discussed throughout this review, several proteases have been targeted and different strategies can and are currently being employed as antiviral therapies, each with their own advantages and disadvantages. We hope that the materials herein collected will give the reader enough insight into current antiviral therapies and protease-inhibition strategies, while also enabling an open discussion on possible future strategies for this ever-growing field.

## Figures and Tables

**Figure 1 viruses-16-00366-f001:**
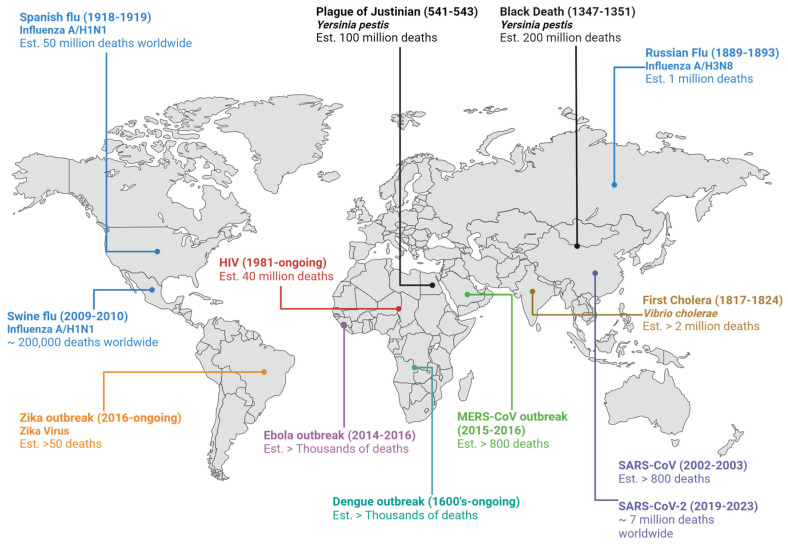
Past pandemics and current outbreaks throughout the world [1,2]. Created with BioRender.com.

**Figure 2 viruses-16-00366-f002:**
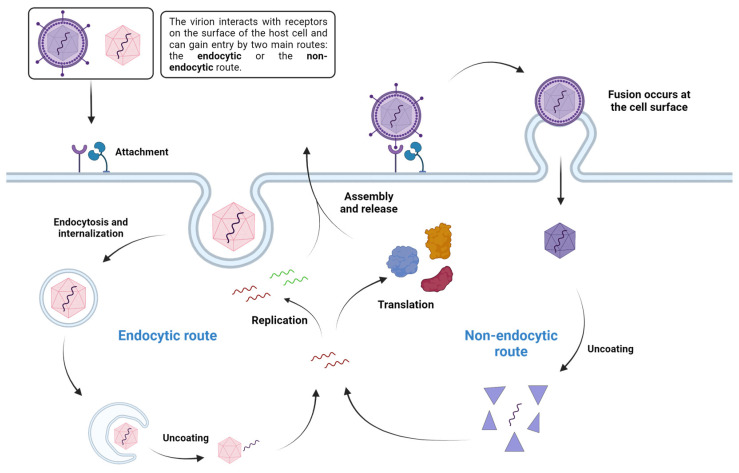
Simplified viral entry and replication mechanism. Created with BioRender.com.

**Figure 3 viruses-16-00366-f003:**
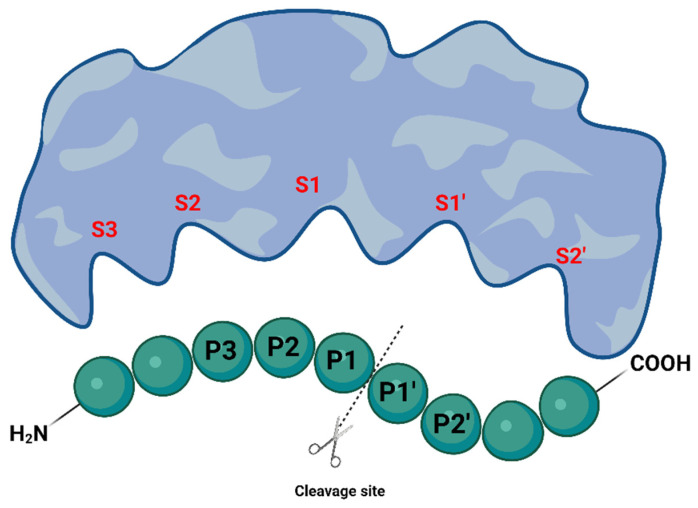
Schechter–Berger nomenclature for substrate recognition by proteases. Created with BioRender.com.

**Figure 4 viruses-16-00366-f004:**
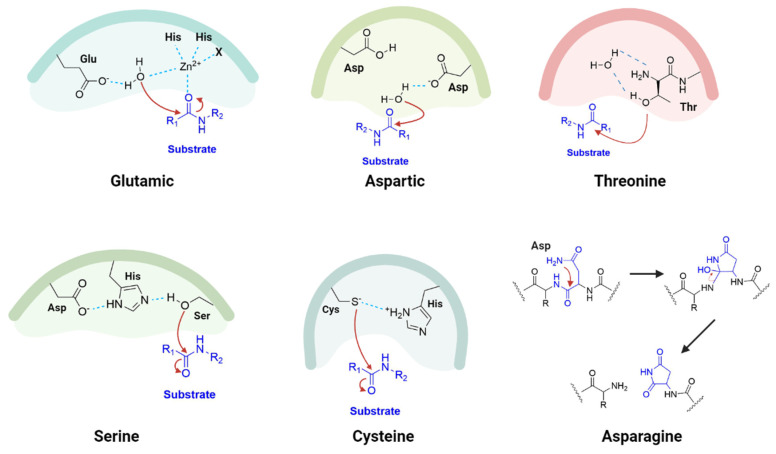
Protease cleavage mechanisms. Created with BioRender.com.

**Figure 5 viruses-16-00366-f005:**
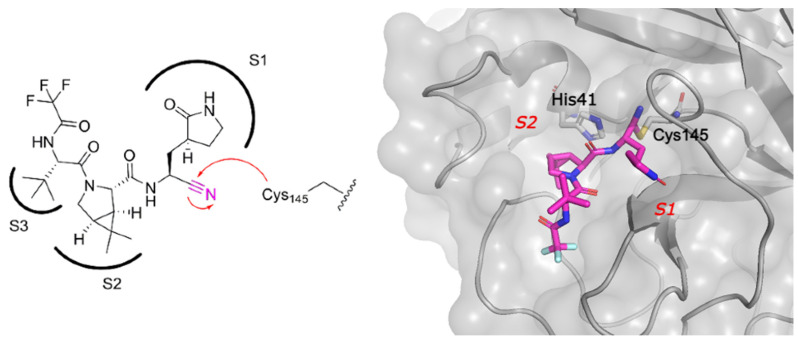
Nirmatrelvir bound to SARS-CoV-2 M^pro^ (PDB ID: 7TE0). The nitrile warhead is depicted in purple. Created with BioRender.com.

**Figure 6 viruses-16-00366-f006:**
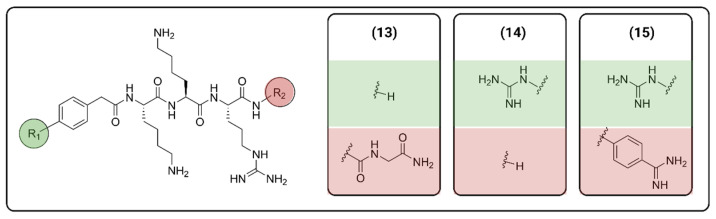
Compounds designed by Phoo et al. [48]. Created with BioRender.com.

**Figure 7 viruses-16-00366-f007:**
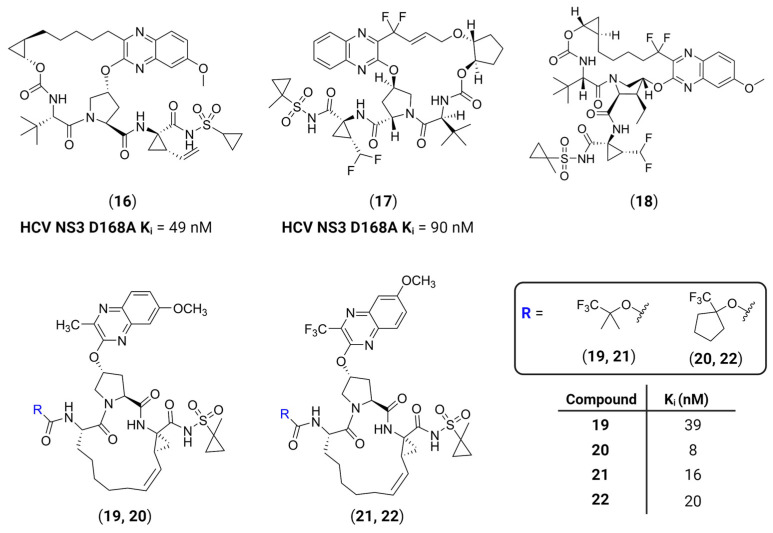
Macrocyclic HCV NS3/4A protease inhibitors and derivatives. R groups depicted in blue are shown in the box on the right. Created with BioRender.com.

**Figure 8 viruses-16-00366-f008:**
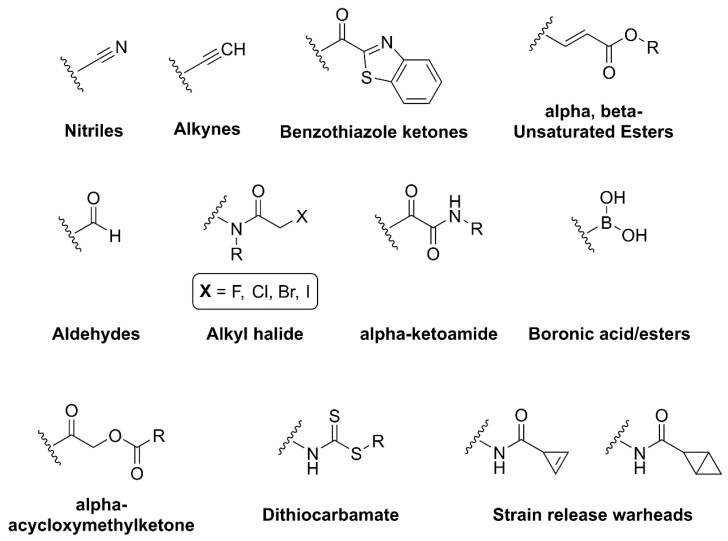
Examples of electrophilic warheads used for targeting proteases. Created with BioRender.com.

**Figure 9 viruses-16-00366-f009:**
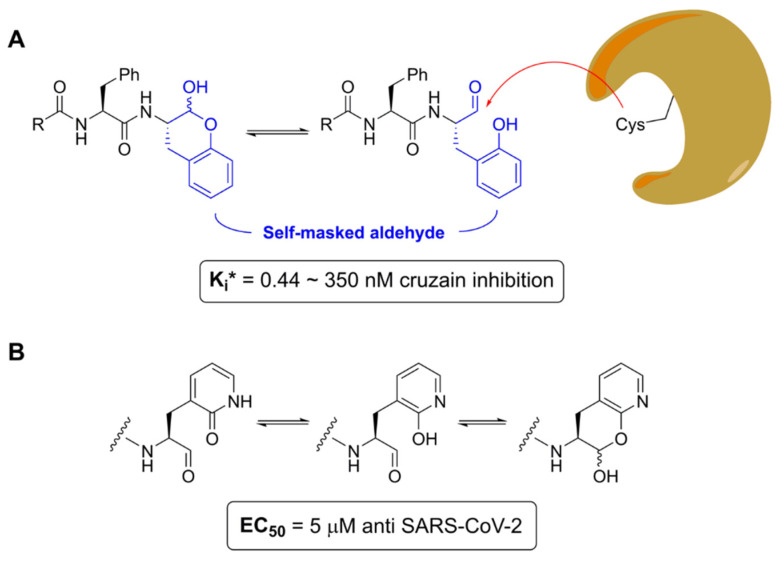
General structure of SMAIs developed by Li et al. [58] (**A**) Masked aldehyde warheads’ (in blue) mechanism of inhibition. (**B**) Bioisosteric replacement of γ-lactam ring via 2-pyridone moiety and anti-SARS-CoV-2 activity. Created with BioRender.com.

**Figure 10 viruses-16-00366-f010:**
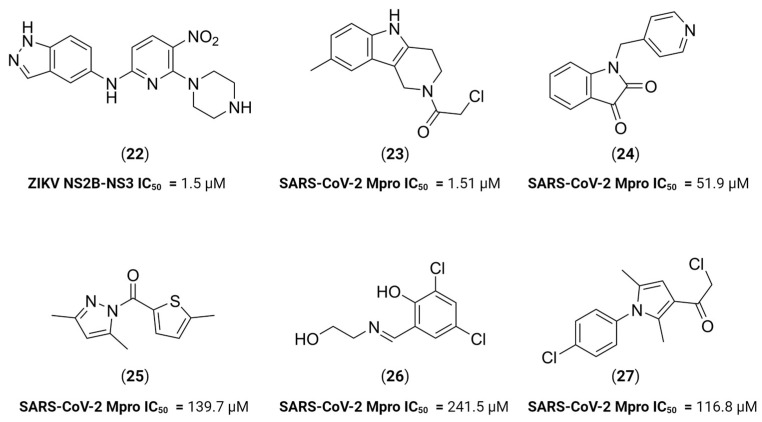
In silico-discovered protease inhibitors. Created with BioRender.com.

**Figure 11 viruses-16-00366-f011:**
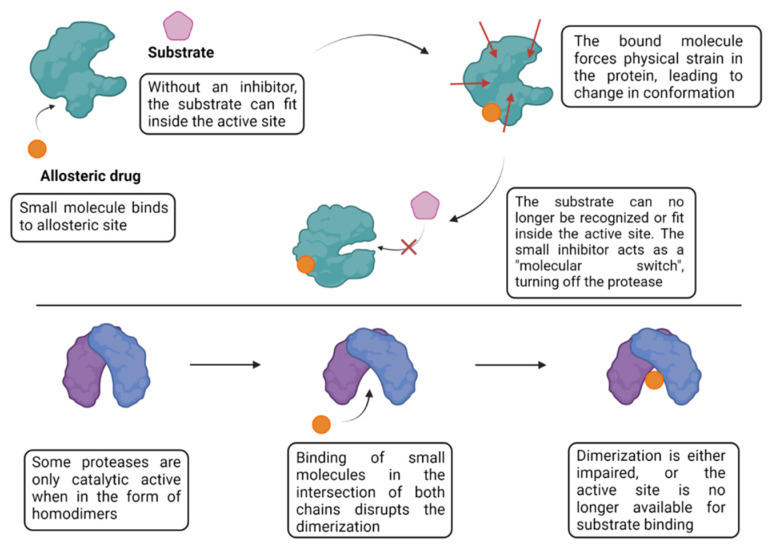
Allosteric inhibition mechanisms by small molecule. Created with BioRender.com.

**Figure 12 viruses-16-00366-f012:**
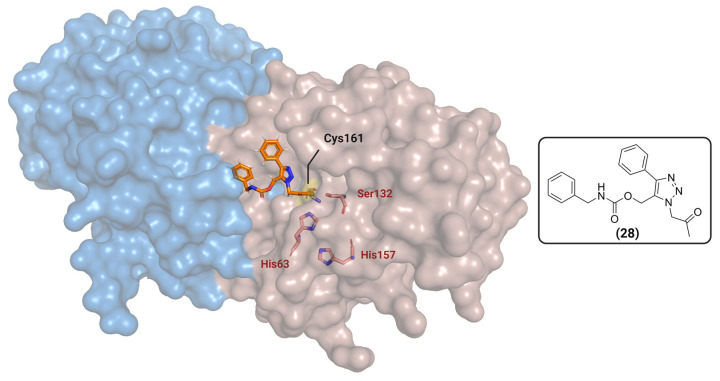
Interaction of aryl triazole derivative (compound 28, in orange) at the dimer interface of the protease (PDB ID: 7TCZ). Each chain is represented in different colors (blue and light red), and the catalytic triad (His63-His157-Ser163) is represented in dark red. Created with BioRender.com.

**Figure 13 viruses-16-00366-f013:**
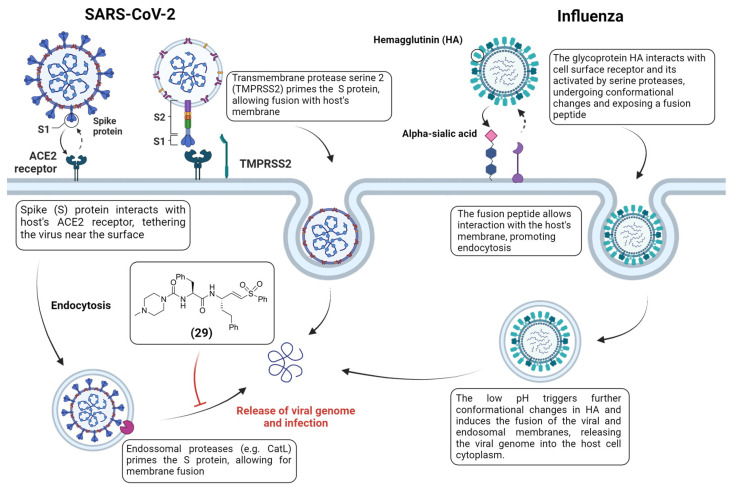
Viral entry mechanisms for SARS-CoV-2 and influenza and the role of a host’s proteases. Created with BioRender.com.

**Figure 14 viruses-16-00366-f014:**
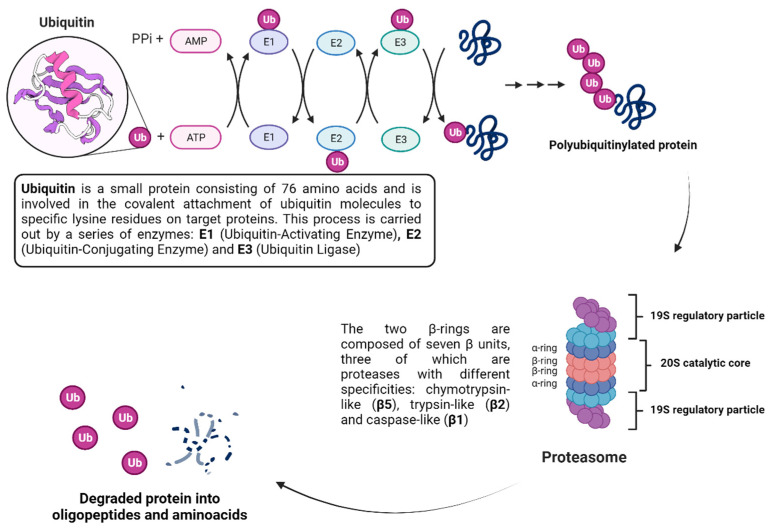
Molecular mechanism of protein ubiquitination and protein degradation by proteasome. Created with BioRender.com.

**Figure 15 viruses-16-00366-f015:**
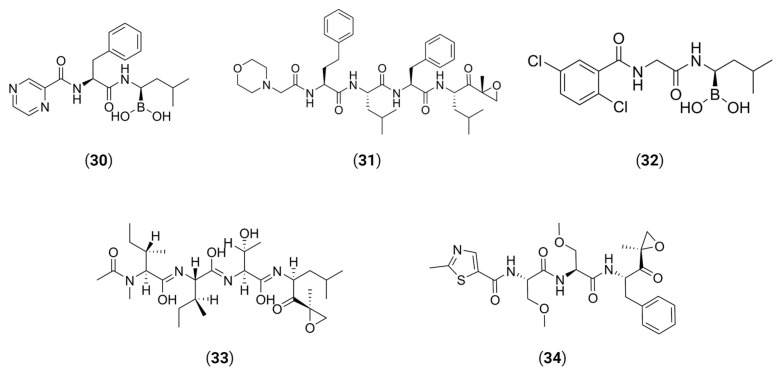
All five proteasome inhibitors were able to inhibit ZIKV and DENV viral replication.

**Figure 16 viruses-16-00366-f016:**
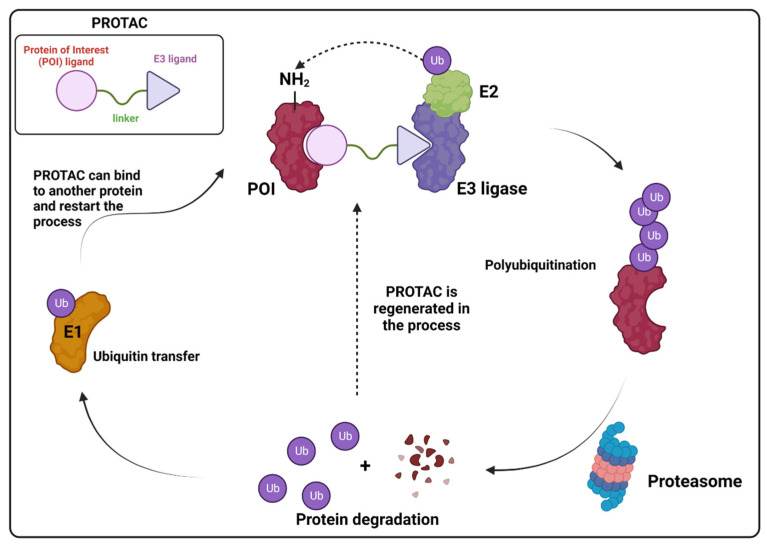
PROTAC mechanism of action. Created with BioRender.com.

**Figure 17 viruses-16-00366-f017:**
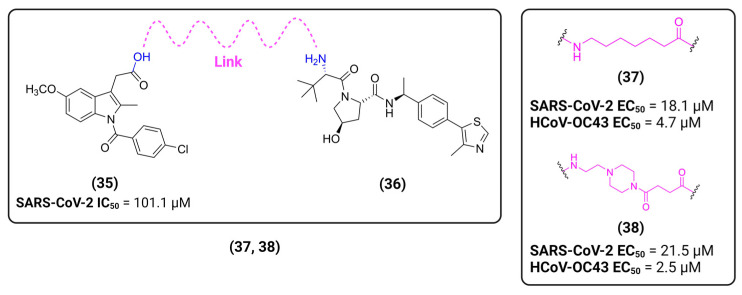
INM-based PROTACs developed by Desantis et al. [148] with broad spectrum activity against coronaviruses. Created with BioRender.com.

**Figure 18 viruses-16-00366-f018:**
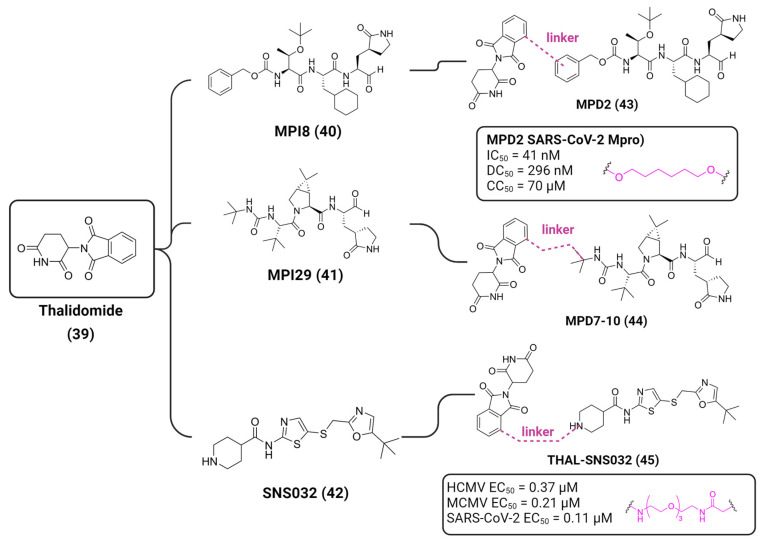
Thalidomide-based PROTACs developed by Alugubelli et al. [149] and Hahn et al. [151] with diverse antiviral activities. Created with BioRender.com.

**Figure 19 viruses-16-00366-f019:**
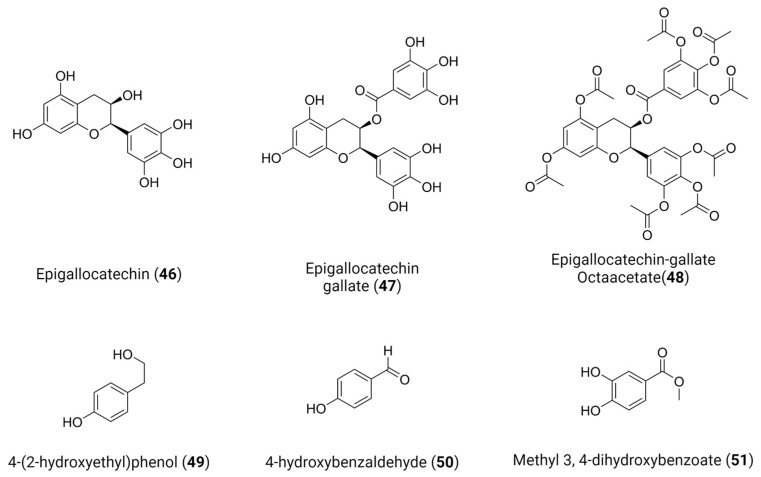
Bioactive compounds with protease activity. Created with BioRender.com.

**Table 1 viruses-16-00366-t001:** Examples of FDA-approved protease inhibitors in the past years *.

#	Structure	Trade Name	Active Ingredient	Year	Indication	Drug Target
**1**	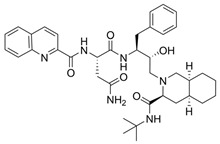	Invirase	Saquinavir mesylate	1995	Anti-HIV drug	HIV-1 protease
**2**	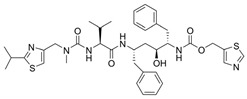	Norvir	Ritonavir	1996	Anti-HIV drug	HIV-1 protease
**3**	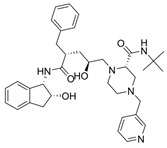	Crixivan	Indinavir	1996	Anti-HIV drug	HIV-1 protease
**4**	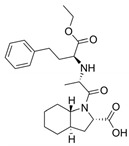	Mavik, Tarka	Trandolapril	1996	Hypertension, congestive heart failure	Angiotensin-converting Enzyme
**5**	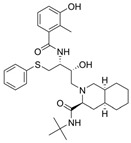	Viracept	Nelfinavir	1997	anti-HIV drug	HIV-1 protease
**6**	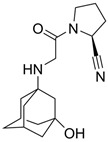	Galvus, Jalra, Xiliarx	Vildagliptin	2008	Antihyperglicemic	Dipeptidyl peptidase-4 (DPP-4)
**7**	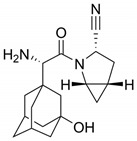	Kombiglyze, Komboglyze, Onglyza, Qtern, Qternmet	Saxagliptin	2009	Antihyperglicemic	Dipeptidyl peptidase-4 (DPP-4)
**8**	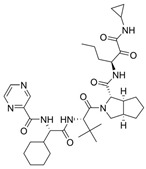	Incivek	Telaprevir	2011	Antiviral, Human Hepatitis C Virus	NS3/4A serine protease
**9**	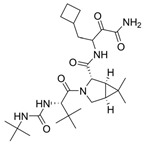	Victrelis	Boceprevir	2011	Antiviral, Human Hepatitis C Virus	NS3/4A serine protease
**10**	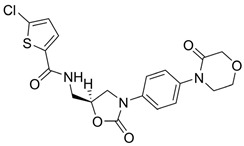	Xarelto	Rivaroxaban	2011	Deep vein Thrombosis, Pulmonary Embolism	Coagulation factor X
**11**	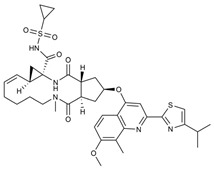	Olysio, Galexos	Simeprevir	2013	Antiviral, Human Hepatitis C Virus	NS3/4A serine protease
**12**	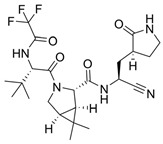	Paxlovid	Nirmatrelvir	2023	Antiviral, SARS-CoV-2	3CLpro cysteine protease (Mpro)

***** All the information was retrieved at the DrugBank database [32].

## Data Availability

Data sharing is not applicable.

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
