# Peer review of "Recent Advances on Targeting Proteases for Antiviral Development"

_viruses, 2024, doi:10.3390/v16030366_

Round 1

Reviewer 1 Report

Comments and Suggestions for Authors

-Figure 1 origin of Spanish flu is not Spain this is an artifact of wartime journalism. What about HIV, or Ebola? What is the criteria for being included on this figure and what is the reasoning behind the locations selected? This figure potentially stigmatizes geographies

-line 52 many host proteins are also necessary for replication, I understand the sentiment, maybe just remove the word "all"

-line 54 name the classes

-line 73 "transcription, replication, and translation" not always in that order

-line 74 what is "synthetic machinery"?

-Figure 2 not every virus enters the nucleus, not every cell has a nucleus

-line 498 please mention more explicitly that k63 ubiquitination is a degradation signal unlike other forms of polyubiquitination

-Figure 13 the ubiquitin is not always conjugated to all the ubiquitination proteins

Comments on the Quality of English Language

Minor editing of English language required

Reviewer 2 Report

Comments and Suggestions for Authors

This review article by Oliveira Borges et al. provides recent advances in development of antiviral drugs targeting proteases. They initially outline viral replication cycle and then introduce how viral protease inhibitors were developed, giving HIV and flaviviruses (e.g., HCV, Zika, Dengue and SARS-CoV2) as examples. The features of this review include the explanation for how lead peptidomimetic compounds were chemically modified for optimization: adding of N-terminal sulfonyl cap (for stability), replacing with bulky moieties (to resistant to cleavage), and coupling with electrophilic warheads (to form irreversible covalent bonds with target proteases). The review provides useful and deep knowledge for virologists. The authors describe allosteric inhibitors and dimerization inhibitors, and also host protease essential for viral replication (e.g., cathepsin, furin, trypsin, and TMPRSS2) as potential drug targets. Finally, they introduce PROTAC, a new technology recruiting E3 ubiquitin ligase through the compound-E3 ligand interactions, leading to target degradation. This application is better than targeting the host proteases, possibly due to less side effects. Overall, this review is comprehensive but well written and interesting. However, there are several points that need to be revised (listed below).

Major:

1) Figure 1: Figure 1 does not appropriately present pandemics/outbreaks in the world. HIV emerged in Africa and spread worldwide (ca. 40 million infected at present).

Dengue burden is higher in Africa than in Americas and highest in Asia (India and southeast Asia).

It is generally accepted that in the 20th century we suffered from three Flu pandemics (Spanish Flu 1918-1919; Asian Flu 1957-1958; Hong Kong Flu 1968-1969).

Zika diverges two lineages: African and Asian lineages. The Zika epidemic was caused by the Asian lineage. It emerged in Malaysia in 1966, spread in Southeast Asia, and then in Polynesia in 2012/2013. It spread from Polynesia to Brazil in 2013. The Zika isolated in Uganda is not an ancestor of the Zika endemic.

What does the size of circles imply (the number of cases)?

Black death and Plaque are both caused by Yersinia pestis bacteria. If you want to include this bacterial infection in the manuscript, it is OK but the colors should be unified (red or black).

The authors need to revise Figure 1 in a balanced fashion.

2) Figure 2: The DNA viruses (except poxviruses) enter the host nucleus and replicate viral genomes. In contrast, the majority of RNA viruses (except influenza and retroviruses) do not enter the nucleus and replicate in the host cytoplasm. The figure needs to be modified, reflecting this point. An alternative would be to erase “Nucleus” from the figure. I want to ask to add "Replication" and point this step in the figure.

3) Dimerization inhibitors: As the authors stated, the first protease inhibitor is Saquinavir that was developed for anti-HIV therapy. Numerous HIV protease inhibitors have been developed and approved by FDA. Since HIV protease forms a symmetric homodimer, a dimerization inhibitor, termed Darunavir, has been developed and clinically used (Chem. Commun., 2022, 58,11762-11782). Darunavir shows a dual mechanism of protease inhibition: block of dimerization (demonstrated by a FRET assay) and block of the catalytic site. I believe that Darunavir is a very good example of protease dimerization inhibitors. The molecular interactions between the compound and HIV protease have been resolved.

4) Macrocyclic inhibitors: In this review, the authors introduce numerous chemical modifications for optimization. However, macrocyclic inhibitors of HCV NS3/4A protease are missing. It would be very helpful to add the information of their chemical modifications, leading to improved specificity and affinity to a shallow catalytic site of NS3.

Minor:

Remove the text messages “Error! Reference source not found” throughout the manuscript.

Reviewer 3 Report

Comments and Suggestions for Authors

The manuscript by P.H.O. Borges et al. is a very comprehensive review on the development of protease inhibitors.  It has a very clear structure and a didactic approach, providing very interesting insights to both new and more advanced students and researchers.  Most illustrations are also very clear, but Figure 2 is an oversimplification of a “universal” viral cycle.  It suggests that all viruses have a nuclear phase, while that is obviously NOT the case for the large class of positive single stranded RNA viruses and also not for some other viruses (e.g. Pox).  That should be corrected.

I understand the pharmaco-development focus of the present review, with most results referring to the very first in silico and in vitro steps.  Nevertheless the reader should also get some information on which of these candidate drugs are progressing into pre-clinical and clinical stages.  Also a short discussion on the potential synergism of this class of drugs with other classes (e.g. polymerase inhibitors and entry inhibitors) would be welcome.  In addition, it should be stressed that the chances on clinical utility of these anti-viral drugs also depend on the acute versus chronic nature of the viral infection.  Again, although these aspects probable do not belong to the core of this review, it is always good to put the more “basic” research into a “translational” context in order to address a wider interested audience and give some connection to the “real world”.        

There is quite a number of small errors in the text, of which the most disturbing is the repeated inclusion (Error! Reference source not found). 

Line 82: play a vital role on several biological: should be IN

Line 87-89: not clear, please rephrase

Line 123: plays a key role on viral: should be in

Line 153-155: while other diseases  have also been benefited by the discovery of inhibitors, such as hypertension, AIDS and  cancer. BETTER: while other diseases, such as hypertension, AIDS and cancer, have also benefitted….

Line 285-186: in low and middle-income countries, such as  Africa, Asia, Brazil and India.   Africa and Asia are continents not countries.

Line 212: are important structural motif. MOTIFS (plural)

Line 217: the presence of the sulfonamide groups have also improved stability: the presence (of the sulfonamides) HAS (= singular)

Line 226: targeting the catalytic residues of proteases have its own challenges: targeting HAS (singular).

Line 227-228: structural changes on the active site: changes IN the active site

Line 228: and resulting in loss of selectivity

Line 527: none of them were able to inhibit the NS2B3 protease: WAS (= singular)

Line 537: PROTACs, or PROteolysis TAgeting Chimeras: “r” omitted in TArgeting

There may be more of these small errors. Please ask a colleague (who is NOT an author and sees the 

Comments on the Quality of English Language

Small errors to be corrected

Round 2

Reviewer 2 Report

Comments and Suggestions for Authors

This review article shows a great improvement through revision by the authors (Oliveira Borges et al). They sincerely revised all the reviewer’s concerns and added discussion about the effectiveness of multidrug therapy, which is now well accepted for HIV, HCV, and tuberculosis.

Reviewer 3 Report

Comments and Suggestions for Authors

The authors took all my remarks into account.